light microscopy/nanotechnology/
materials science

nanodiamond, super-resolution, multi-photon
excitation, adaptive optics

**Author for correspondence:**
Brian R. Patton
e-mail: brian.patton@strath.ac.uk

# Nanodiamonds enable adaptive-optics enhanced, super-resolution, two-photon excitation microscopy

Graeme E. Johnstone, Gemma S. Cairns
and Brian R. Patton

Department of Physics and SUPA, University of Strathclyde, Glasgow G4 0NG, UK

 GEJ, 0000-0001-5471-4664; GSC, 0000-0003-0524-8008;
BRP, 0000-0001-8222-4419

Particles of diamond in the 5–100 nm size range, known as nanodiamond (ND), have shown promise as robust fluorophores for optical imaging. We demonstrate here that, due to their photostability, they are not only suitable for two-photon imaging, but also allow significant resolution enhancement when combined with computational super-resolution techniques. We observe a resolution of 42.5 nm when processing two-photon images with the Super-Resolution Radial Fluctuations algorithm. We show manipulation of the point-spread function of the microscope using adaptive optics. This demonstrates how the photostability of ND can also be of use when characterizing adaptive optics technologies or testing the resilience of super-resolution or aberration correction algorithms.

## 1. Introduction

The biocompatibility of diamond [1], combined with the ability to make nanoscopic particles of less than 100 nm diameter, has led to research into the use of nanodiamond (ND) for a variety of biological applications [2], including drug delivery [3,4] and use as a fluorescent marker for microscopy [5,6]. The surface chemistry of ND allows it to be functionalized by chemically attaching a range of different molecules to the ND surface [7,8]. Examples of functionalization strategies include antibody [9,10] and DNA [11] binding. The benefits arise by allowing efficient targeting of ND to subcellular features of interest, thereby allowing effective labelling for fluorescent microscopy [2].

The fluorescence of ND is due to defects in the diamond structure [12]. A large number of defects are known to show fluorescent properties [8], but of particular interest is the nitrogen-vacancy (NV) defect, which occurs when two

neighbouring carbon atoms are replaced by a nitrogen atom and a vacant space in the crystal lattice. This defect is most commonly found in a negatively charged state, referred to as $NV^-$, and is efficiently imaged with a single photon excitation of 532 nm that results in emission over a wide wavelength band from 637 nm to approximately 800 nm. A particular advantage of $NV^-$ emission is that, unlike most conventional fluorophores used for microscopy, it is completely photostable, allowing imaging of individual cells labelled with ND over periods of a week or more [13]. The brightness of an individual ND is a function of the number of defects it contains. This number can be increased during the production of ND, allowing for even brighter labelling, and is typically correlated with the size of the ND: particles smaller than 5–10 nm are unlikely to be able to support a fluorescently active NV [14,15] without specific processing [16]. By contrast, 100 nm particles may have hundreds of active NV centres. This flexibility allows a trade-off between the intrinsic brightness of the ND (improving signal to noise for a given image acquisition rate) and the potential impact of the ND on the local biological processes (due either to physical blocking or surface chemistry effects). There have been many studies demonstrating the use of ND with standard microscopy techniques, including confocal microscopy [17] and super-resolution imaging in the form of stimulated emission depletion microscopy [18,19].

An additional driver for the interest in NV as a fluorophore relates to the quantum mechanical properties that can be accessed optically [20]. By combining optical excitation and readout (as is performed when imaging in a microscope) with some pulsed microwave ($\approx$2.88 GHz) electronic-spin manipulation, it is possible to use the $NV^-$ centre as a sensor that can detect a number of features of the local environment, such as the temperature [6,21] and the presence of magnetic fields [22–25]. Active sensing of environmental changes induced by biological processes at a sub-micrometre scale, with NV centres, would extend the usefulness of ND in a range of imaging applications. However, while the emission of $NV^-$ is in a wavelength range with low scattering and absorption by water, the single-photon excitation wavelength of 532 nm poses problems related both to scattering and autofluorescence effects that will decrease the effective sensitivity of the ND to the signals of interest.

One alternative approach is to take advantage of two-photon excitation (TPE) microscopy, which has the ability to image at a greater depth than single-photon microscopy [26] due to decreased scattering and absorption. There is also a better signal-to-noise ratio as the fluorescence is only generated at the excitation focus and not throughout the whole illumination cone, giving significant benefits in autofluorescent samples. It has recently been shown that NV centres in bulk diamond samples can fluoresce with TPE [27] and likewise for NV in fragments of diamond of sizes 10–100 µm [28]. Tuning the excitation wavelengths in a range from 1030 to 1310 nm allows preferential excitation of $NV^-$ or the neutral NV0 state. There have been initial reports of TPE fluoresence in NDs [29–31], showing that they are compatible with this mode of imaging.

In this paper, we further demonstrate TPE imaging of NV centres in ND and show how ND is particularly suited to computational super-resolution techniques, such as that enabled by the super-resolution radial fluctuations (SRRF) [32] approach. To allow high-quality super-resolution imaging, we take advantage of the adaptive optics technology incorporated in our microscope. By using deformable mirrors, which can introduce controllable distortions to the wavefront of the light that propagates through the microscope, it is possible to compensate for the optical aberrations introduced by the inhomogeneous nature of the samples we wish to image. For further understanding of the requirements for implementation of adaptive optics within microscopy, we recommend [33,34]. While multi-photon microscopy is inherently suited to imaging in optically aberrating samples, it nevertheless benefits from adaptive optical image correction [35–37]. With this in mind, we also demonstrate adaptive optic control of the excitation focal volume and the resulting changes in the images obtained from single ND crystals. By combining adaptive-optics and computational super-resolution imaging, we also show that ND is a superb fluorophore with significant potential for applications in which efficient correction of aberrations deep within an aberrating medium (such as tissue) is essential.

# 2. Experimental set-up

## 2.1. Materials and sample preparation

The NDs used in these experiments were produced by Adamas Nanotechnologies. We prepared slides for imaging from a mix of two monodisperse suspensions (both 0.1%w/v) of 40 and 100 nm diameter ND. The 40 nm ND each contain approximately 10 NV while there are closer to 400 NV per 100 nm ND as per the manufacturer's calibration information. To prepare a suspension suitable for deposition on a coverslip, we first sonicated each source of NDs to break up larger aggregates before adding

10 µl of each ND suspension to 100 µl of distilled water. The resulting suspension was then deposited onto a #1.5 microscope glass coverslip and allowed to dry to ensure some ND adhered to the coverslip before being mounted on to a microscope slide using a small amount of distilled water as a mountant medium and finally sealing the sample with nail polish.

## 2.2. Microscope and imaging equipment

The microscope that provided all of the imaging for this work is a custom-designed, confocal microscope. A 1.35NA oil immersion objective lens was used for imaging in an epi-flourescence geometry. We used two excitation lasers for the work presented in this paper:

— A less than 50 ps pulse-duration 532 nm Picoquant laser running at a repetition rate of 40 MHz. This laser was used for single-photon excitation of $NV^-$ centres.
— A Coherent Fidelity 2 pulsed fibre laser with a wavelength centred on 1070 nm, a bandwidth of 70 nm and a pulse duration of 40 fs at the laser's output. A chirp compensator allows optimization of the pulse duration at the sample by maximizing the observed multi-photon signal.

The fluorescent light emitted from the sample was coupled via an optical fibre (acting as the confocal pin hole for the single photon excitation) to a single pixel detector, a Laser Components avalanche photodiode (Count-50). The final lens before the optical fibre was chosen to set an effective pinhole size of 1 Airy unit. This may give a slight increase in contrast in two-photon imaging mode, however the effect of the pinhole will be dominated by the inherent optical sectioning provided by the TPE process. Images were created by raster scanning the beam across the sample using a Newport Fast Steering Mirror (FSM-300) and axial-imaging was performed by moving the objective which was mounted on a Physik Instrumente piezo positioner (P-725 PIFOC). We have chosen the detection path to have a wavelength sensitivity of 650–750 nm. This range corresponds with the peak of the emission from $NV^-$ centres. As will be discussed further in §4, a Boston Micromachines Corporation deformable mirror (Multi-DM) with 140 actuators and 3.5 µm stroke is used for aberration correction of both the excitation and signal paths. Control of the microscope was performed by custom-written Labview software driving a National Instruments FPGA.

# 3. Two-photon imaging

We begin by demonstrating the effectiveness of two-photon imaging with our system. Figure 1 shows a comparison of one-photon and two-photon imaging on our mixed ND sample. We processed the data initially in Fiji [38,39] and then generated the output figures for publication in OriginPro. We present the images using 'CubeHelix' [40], a colourmap that increases linearly in perceived brightness, thereby making it suitable for colour-blind readers and reproduction in greyscale media. All unprocessed imaging data is available at the DOI given in [41] along with descriptions of the file format generated by the custom software running our microscope and scripts for importing the data into Fiji.

Figure 1 shows the comparison between single-photon and multi-photon imaging of a single region of our sample. Figure 1a,c shows that there is excellent correlation between the NDs that appear when imaging with 532 nm excitation and those present with 1070 nm excitation, respectively. To show the comparable resolution in both cases, figure 1b,d shows a higher resolution scan of the same pair of NDs within the larger scan.

An advantage of multi-photon excitation is that it provides intrinsic sectioning when imaging, due to the excitation only occurring in the central volume of the excitation point-spread function (PSF). We include a video (electronic supplementary material, S1), which shows multiple NDs at different z-positions within the sample when imaged with the 1070 nm excitation.

While figure 1 demonstrates that we are seeing multi-photon excitation of the NV within our ND, we also wanted to confirm that it is a two-photon process. To confirm this, the beam was centred on a single ND and the detected photon counts were measured versus excitation power for both single- and multi-photon excitation modes. The results are shown in figure 2 for the 532 nm excitation and in figure 3 for the 1070 nm excitation. In the 532 nm excitation case, and at low input powers, the number of counts increases linearly with the power of the excitation laser. This is as expected for single-photon fluorescence. For all fluorophores, as the excitation power increases further, the emitted fluorescence increases sub-linearly and starts to plateau—it saturates [42]. Figure 2 shows exactly this behaviour in the single-photon excitation of ND, with saturation already apparent at 100 µW incident power and an

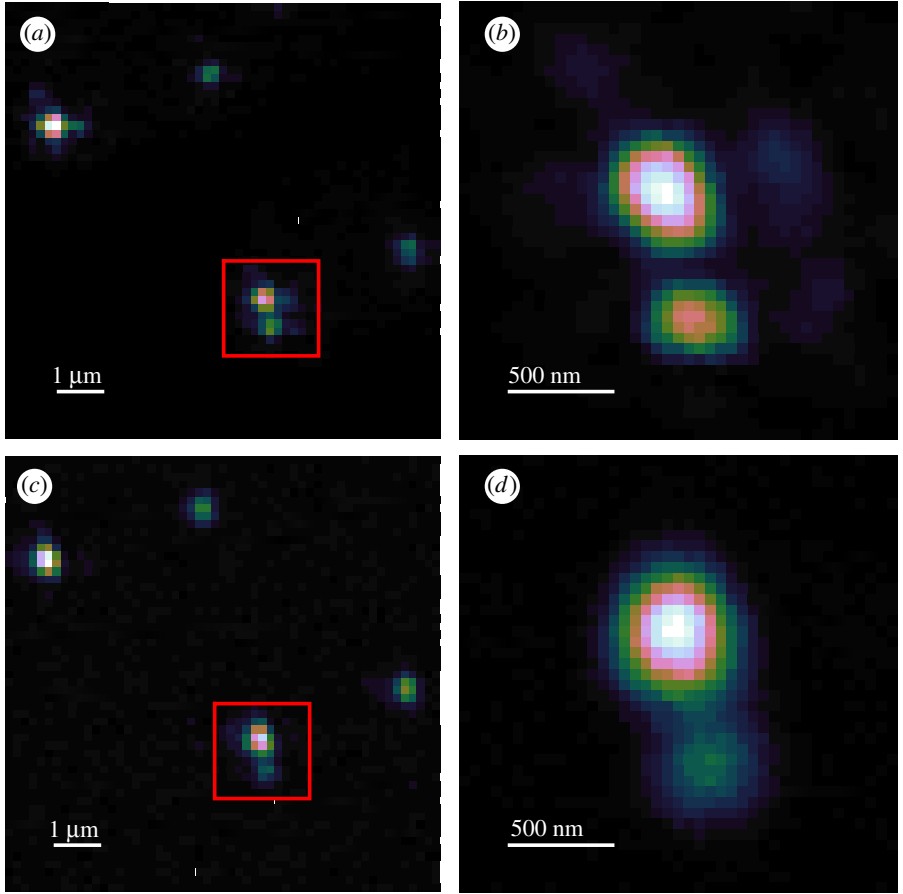

**Figure 1.** (*a*) Cluster of NDs imaged with single-photon excitation confocal microscopy using 532 nm laser excitation. (*b*) A higher resolution 532 nm excitation scan of the region highlighted in (*a*). (*c*) The same region as (*a*), imaged in two-photon mode with 1070 nm excitation. (*d*) A higher resolution 1070 nm excitation scan of the region highlighted in (*c*). The sample contains a mixture of 40 and 100 nm diameter NDs. The colour scale for all images is linear and normalized to the maximum and minimum pixel values of each individual image.

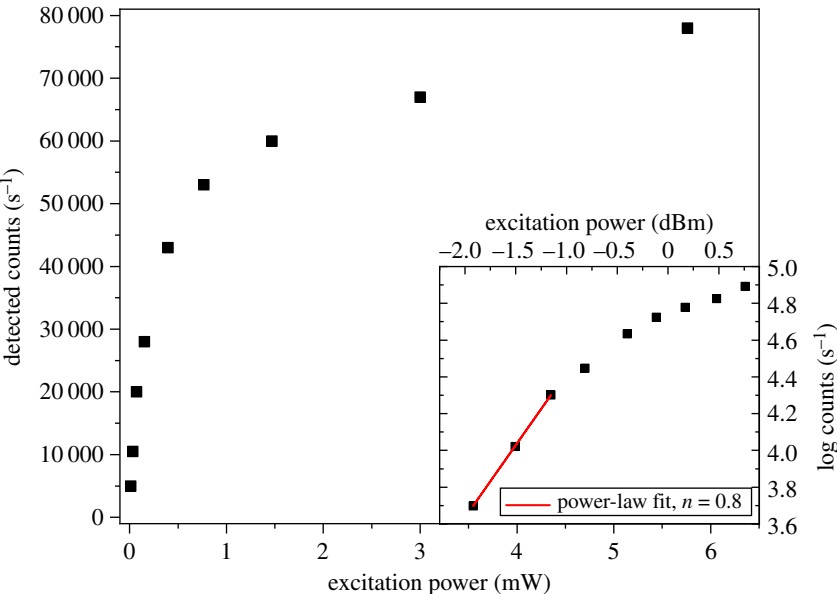

**Figure 2.** Observed fluorescence count rate versus excitation power for single-photon excitation with 532 nm laser. Inset: Logarithmic power dependence showing sub-linear power-law behaviour even at low excitation intensities. Scale is 1 dBm = 1 mW as measured at sample.

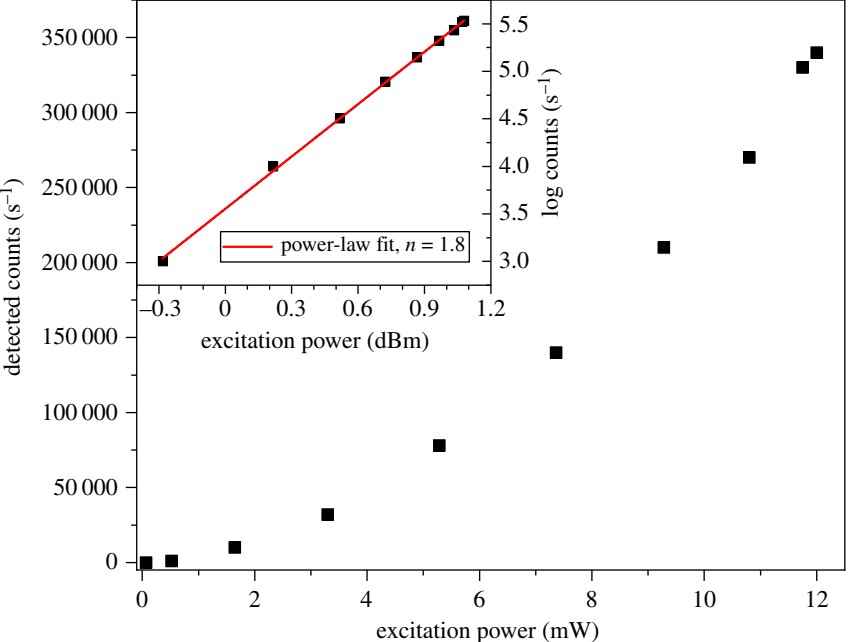

**Figure 3.** Observed fluorescence count rate versus excitation power for multi-photon excitation with 1070 nm laser. Inset: Logarithmic power dependence showing approximately quadratic power law behaviour (exponent $n = 1.8$) indicating two-photon excitation. Scale is 1 dBm = 1 mW as measured at sample.

estimated peak count rate of approximately $80\,000\,\mathrm{s}^{-1}$. The inset of figure 2, plotting count rate and power on logarithmic scales, shows a power-law fit with power parameter $n = 0.8$ at low excitation power. That the value of $n$ is less than one likely reflects the fact that we included points where saturation is already beginning to occur. It is worth noting here that it is often undesirable to image fluorophores in the saturation regime as it increases the probability of a photo-bleaching event due to inter-system crossings when the fluorophore is in the excited state. However, the photostability of NV centres means that it is possible to image them indefinitely even when well into the saturation regime. This is particularly relevant when seeking to use the emission properties of the NV as an environmental sensor. Knowing the relationship between excitation power and degree of saturation is useful as it allows us to maximize the photons being detected while decreasing the possibility of other phototoxicity effects from the excitation laser.

In the case of the 1070 nm excitation, figure 3, we see a super-linear dependence of the emission on the excitation power. For a pure two-photon process the count rate should be quadratic with the excitation intensity. Performing a power law fit we recover a power parameter of $n = 1.84 \pm 0.02$ indicating that we are seeing TPE as the mechanism for generating fluorescence at this excitation wavelength. There are two phenomena here that are worth commenting on. Firstly, we do not see saturation in the TPE power dependence. As discussed in the description of the microscope, we are limited to 12 mW power at the sample for the 1070 nm excitation laser, so this is likely a result of insufficient available laser power. The second point of note is that the TPE appears to show higher fluorescence counts than the single-photon excitation at saturation. We do not currently have a full explanation for this. One contribution to the observed count rates comes from the fact that the 1070 nm laser was operated at a higher pulse repetition rate than the 532 nm laser (70 versus 40 MHz). For short-duration laser pulses ($\tau_{\mathrm{Pulse}} < \tau_{\mathrm{NV}^-}$), this is equivalent to stating that each pulse results in one photon being emitted. Therefore, for repetition rates $< 1/\tau_{\mathrm{NV}^-}$, as is the case here, we would expect the count rate to be proportional to the laser repetition rate. Ji *et al*. [28] also shows that, even with the 650 nm long pass filter in our system, there can still be significant NV0 emission leaking into the NV$^-$ channel. Furthermore, with 1070 nm excitation there is a significant cross-section for NV0 excitation. We did not have appropriate spectroscopic equipment to unambiguously determine the contribution of the NV0 to this excess count rate. However, we note that this would mainly be an issue for the use of the NV$^-$ spin properties in local field sensing. For general two-photon imaging, the apparent brightness of the ND with the excitation wavelength and power demonstrated here is beneficial. Should the application require efficient NV$^-$ only excitation, then Ji *et al*. [28] suggests longer wavelength excitation (more than 1150 nm) will improve the emission from NV$^-$.

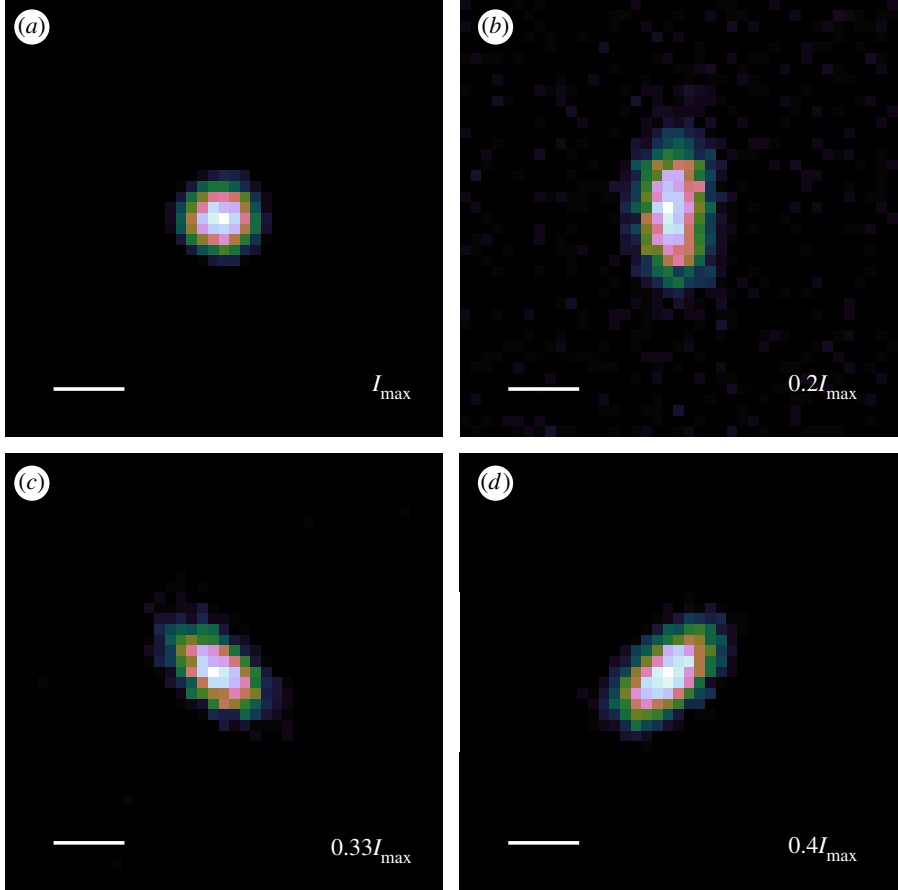

**Figure 4.** Experimental demonstration of deformable-mirror based control of the two-photon point-spread function (PSF). (*a*) Optimally corrected PSF. Aberrated with: (*b*) vertical astigmatism, (*c*) positive oblique astigmatism, (*d*) negative oblique astigmatism. All scale bars are 500 nm, the colourscale is linear and ranges down from the given maximum pixel value (shown relative to the peak of *a*, the brightest image) to 10% of that value for each sub-figure.

# 4. Adaptive optics

Typical implementations of adaptive optics in two-photon microscopy [35] use active elements in the excitation path only and, due to the signal being generated only at the focal volume, employ wavefront sensors to directly detect the aberrations. As we are using a confocal detection scheme, the microscope used for these measurements has a deformable mirror (DM) installed in the common beam path and in such a way that it is imaged onto the back aperture of the objective. Deformations of the mirror can therefore be used to implement adaptive optical control of the PSF, with particular emphasis on correcting aberrations induced by the sample and which degrade image quality. Orthogonal control modes for the DM are derived from the individual actuator influence functions as described in references [43,44] and we use a sensorless correction procedure [36,45] in which image-based metrics are used to measure the optimal shape of the DM. A driver for the development of efficient AO correction schemes has been the desire to minimize the photons lost to the correction process; if the sample undergoes significant photobleaching during the aberration estimation then it may call into question the use of AO in that particular sample. The lack of photobleaching of the NV centre, therefore, makes it an ideal candidate for implementing and testing aberration correction techniques.

In figure 4, we demonstrate control over the PSF of our two-photon confocal imaging system. All sub-figures are of the same ND, differing in the abberation applied to the DM and with a colour scale normalized to the maximum pixel value shown in the sub figure. Since aberrations are expected to decrease the peak intensity of the PSF, we also show this peak value as referenced to the peak intensity of the unaberrated PSF (figure 4*a*). For this demonstration, we have applied modes corresponding to the two primary astigmatism aberrations (Zernike modes 5 and 6, using Noll indices). Figure 4*b* shows vertical astigmatism, while figure 4*c,d* shows positive and negative amounts

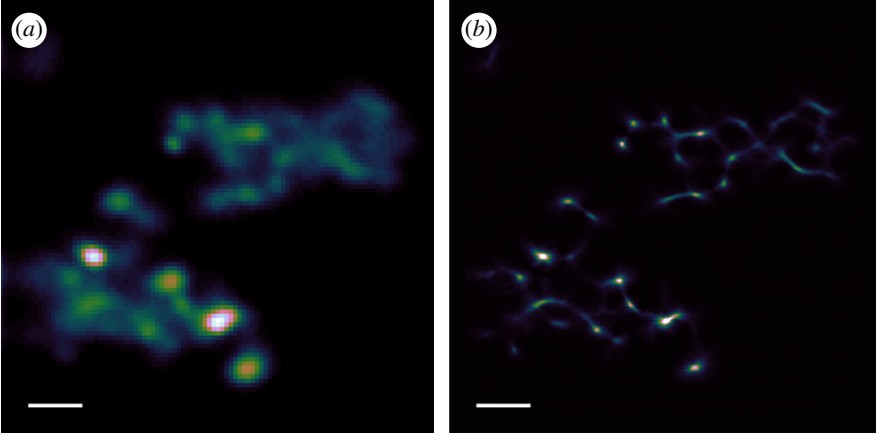

**Figure 5.** Super-Resolution Radial Fluctuations (SRRF) imaging of two-photon excitation ND samples. (*a*) Drift-corrected sum of 100 acquisitions in TPE acquisition mode. (*b*) Super-resolution image derived from original image stack. Scale bar 1 μm, linear colour scale applied to each image.

of oblique astigmatism. The fidelity with the expected PSF shape is good, with little cross-talk between modes. The combination of two-photon excitation microscopy, ND as a fluorophore and adaptive optics will allow deeper imaging in tissue than is possible with either technique alone and we look forward to further exploring the possibilities in future work.

# 5. Superresolution imaging of two-photon excitation microscopy

The photostability of the NV defect makes it particularly suited to computational super-resolution techniques such as Super-Resolution Radial Fluctuations (SRRF) imaging. The algorithm and theory underlying the method is described in [32], and a plug-in module for ImageJ makes it a particularly user-friendly technique. SRRF makes use of the radial symmetry of the PSF to generate a radiality map that locates the fluorophore within the image. By then taking advantage of correlations in naturally occurring fluctuations in the radial symmetry measured over a large number of images of the same object, the SRRF algorithm can extract super-resolution images. In practice, a minimum of 100 standard resolution images are synthesized into a single SRRF image. It is worth noting that current versions of the SRRF algorithm do not offer any resolution enhancement along the optical axis.

Figure 5*a* shows a pair of ND clusters imaged using two-photon excitation microscopy. This image was generated by repeating 100 times and processed by correcting for in-plane sample drift, using the NanoJ-Core algorithm [46], followed by summing the 100 frames. With a 1.5 ms pixel dwell time, and an image size of 101 × 101 pixels, each frame takes approximately 18 s to complete (including line flyback time), with the full image stack therefore taking approximately 30 min. To generate the SRRF image shown in figure 5*b*, we again applied the drift-correction routine before using the NanoJ-SRRF plugin with settings: Ring Radius = 0.5, Radiality Magnification = 6, Axes in Ring = 7 and Temporal Radiality Average. The resultant image shows significant increase in resolution, however there are some artefacts present; along with the lines joining individual NDs, inspection of the raw data shows predominant sample drift along the Z-axis and the presence of two distinct layers of ND separated in the Z-axis. From the drift correction table generated by NanoJ, which corrects only for lateral drift, we see this corroborated with a maximum lateral drift of 45 nm. Electronic supplementary material, Movie S1 shows the results of a three-dimensional stack of this cluster.

To better demonstrate the quality of SRRF imaging possible with ND, we performed further measurements on the upper cluster of ND imaged in figure 5. By keeping the pixel size approximately the same (70 nm versus 80 nm in figure 5) but decreasing the image boundaries, we hoped that the resulting shorter scan times would decrease the sample drift along the optical axis. Drift in this axis cannot be efficiently corrected on our hardware and has a significant deleterious effect on the SRRF processing algorithm. The results of these measurements are shown in figure 6, with the 100 drift-corrected and summed standard resolution TPE frames shown in figure 6*a* and the SRRF processed image in figure 6*b*. In this case, the frame time is about 5 s and while the maximal lateral drift as determined by the drift correction routine is now 63 nm, there is much less Z-axis drift

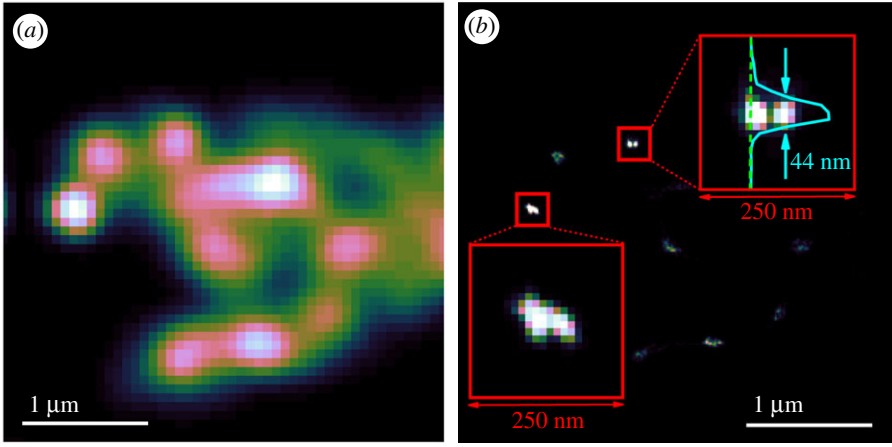

**Figure 6.** Higher frame-rate imaging of upper cluster of ND from figure 5 (*a*) drift-corrected sum of 100 acquisitions in TPE acquisition mode (*b*) super-resolution image derived from original image stack. Insets are from two, 250 nm$^2$ regions as highlighted on main SRFF image. Upper inset also shows intensity plot along a vertical slice through one of the SRFF-resolved emitters. Indicated is the 44 nm full-width-half-maximum of a Gaussian fit to this intensity profile. 1 μm scale bars refer to main images, linear colour scale applied to each image.

apparent in the raw data. As is the case with the larger clusters in figure 5, the SRRF algorithm is able to recover significant extra resolution, and the higher quality initial imaging translates into higher SRRF resolutions. We will return to this point below. To show the enhancement in resolution, figure 6*b* includes insets that enlarge two 250 nm$^2$ regions of interest. In the lower region, we show that there is structure now visible within the brightest diffraction-limited spot within the TPE image of figure 6*a*. The lateral extent of this emitting region is of the order of 100 nm, however, we do not have sufficient information to know unambiguously whether we are observing structured emission from a single 100 nm ND or emission from a small cluster of 40 nm ND. The upper region highlighted on figure 6*b* appears to be clearer in interpretation; there are two closely spaced emitters that the SRRF map shows as being distinguishable. To gain an initial estimation of the resolution of the SRRF image, we show a vertical slice through the left emitter of the pair along the green dashed line. Fitting this intensity profile to a Gaussian gives a 44 nm full-width-half-maximum (FWHM), consistent with this being a single 40 nm ND.

We have shown that sample drift can adversely affect the quality of an SRRF reconstruction. An advantage of our microscope, when combined with the lack of photobleaching of the ND, is that we can also explore the effects of optical aberrations on the SRRF process. Figure 7 shows a demonstration of how this can be put into practice. As a reference, the upper row of figure 7*a,b* shows the unaberrated TPE and SRRF images. We then acquired further TPE image sets with identical imaging conditions apart from a known aberration being applied to the DM. For the middle row (figure 7*c,d*), we applied 1 unit each of astigmatism and coma (Zernike Noll indices 6 and 7). To further quantify the effect of the aberration, we consider the effective RMS wavefront error, $\sigma$, being applied by the DM. In this case $\sigma = 1.10$ rad. From this, we can also estimate the Strehl ratio as $S = \mathrm{e}^{-\sigma^2}$. The Strehl ratio gives an indication of image quality, with $S = 1$ representing perfect (unaberrated) imaging conditions and decreasing values of $S$ corresponding to increasing image degradation due to aberrations. For general imaging $S < 0.8$ is considered to represent poor imaging conditions. For $\sigma = 1.10$ rad, this results in a Strehl ratio of 0.30. This is comparable to the Strehl ratio previously observed [47] when imaging at a depth of 10 μm in cleared *Drosophila melanogaster* brain tissue. The resulting SRRF images show more noise, no clear structure in the features highlighted in figure 6*b* and artefacts that appear to link previously distinct features. To demonstrate the effects of even higher aberrations, $\sigma = 1.43$, $S = 0.12$, we applied three units each of coma and astigmatism, with the results being shown in figure 7*e,f*. While SRRF still recovers some features with this level of aberration, the image is significantly degraded when compared to either of the other two aberration cases. It is worth noting that the Strehl ratio can also be calculated as the ratio of the peak aberrated intensity to the peak unaberrated intensity. From the peak intensities in figure 7*a,c,e*, and assuming that *a* is a reasonable approximation of the no-aberration case, we find that the measured low aberration Strehl ratio is 0.54, while the high aberration is 0.13. We attribute the difference between the expected Strehl ratios and the empirical ones to be due to drift in the calibration of the deformable

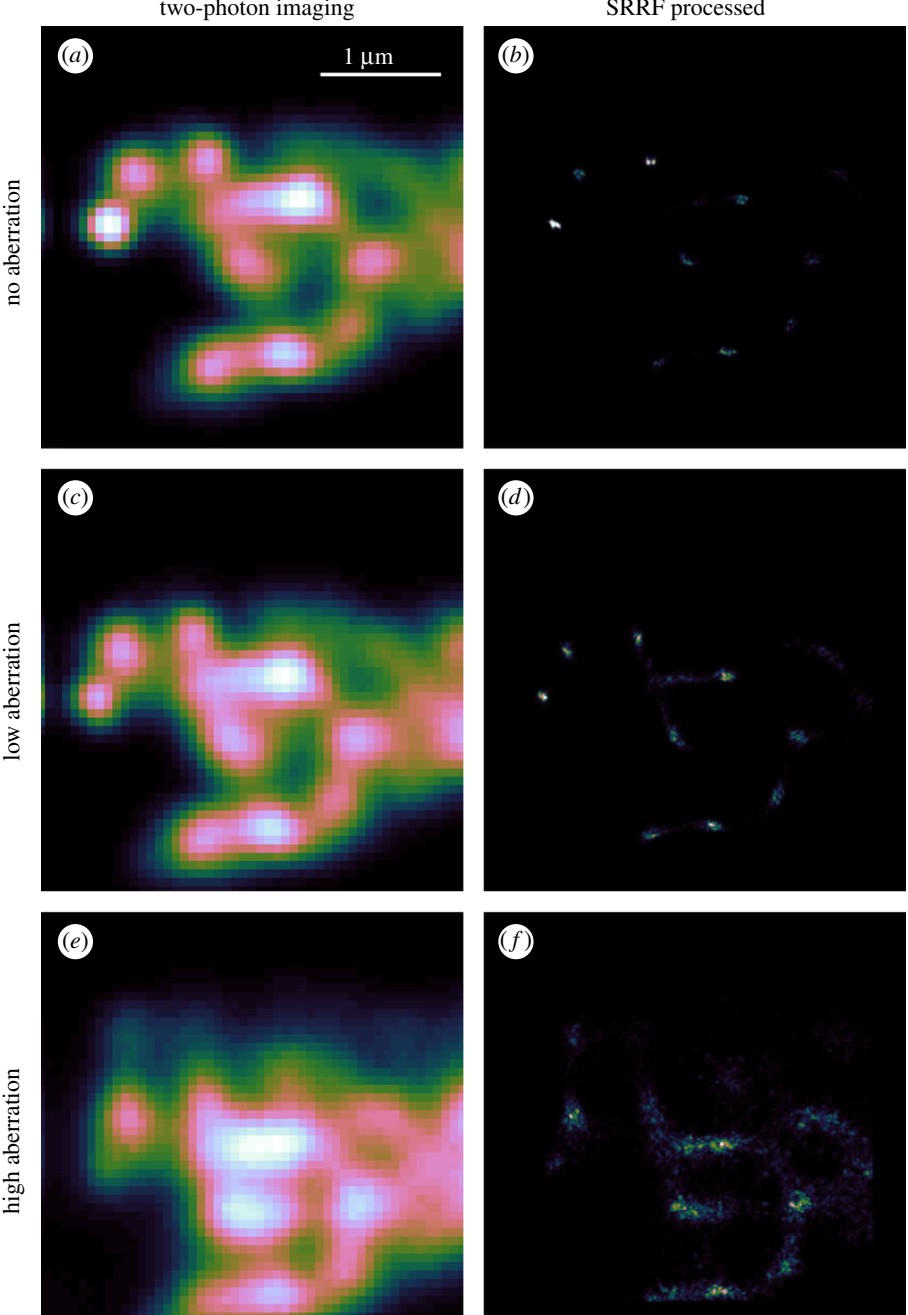

**Figure 7.** Demonstration of the effects of aberration on TPE and SRRF imaging. (a, c, e) Drift-corrected sum of 100 TPE frames with no ((a, b) peak counts 92 000), Low ((c, d) peak counts 50 000) and High ((e, f) peak counts 12 000) amounts of applied aberration. (b, d, f) Corresponding SRRF images. All images at same scale, with 1 μm bar shown in a for reference. Linear colour scale applied to each image.

mirror. While figure 4 shows that the applied modes behave according to their expected aberration class, we did not obtain sufficient quantitative information to confirm the modal amplitude calibration of the DM. Once more, we would like to emphasize that the photostability of the ND emission allows us to obtain high-quality and trustworthy data, such as that being discussed in this paragraph, that is invaluable in the alignment and characterization of complex optical microscopy experiments.

To better quantify the effective resolution of the SRRF images in all three cases shown in figure 7, we applied the Fourier Ring Correlation technique (FRC) [48]. This works on a pair of images that differ only in noise: at low spatial frequencies which are dominated by the (identical) structure of the sample there is a high degree, FRC ≈ 1, of correlation, which decreases with increasing spatial frequency until reaching a point at which the images are dominated entirely by uncorrelated noise. The spatial frequency at which this crossover occurs, often referred to as the FRC resolution, is a measure of the maximum spatial

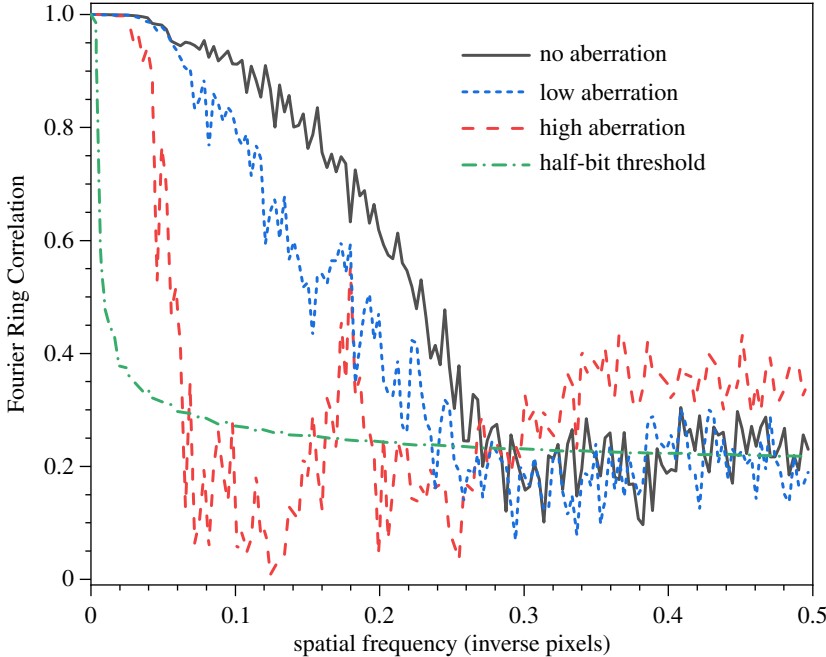

**Figure 8.** Fourier Ring Correlation curves for SRRF data in figure 7. Horizontal scaling is inverse pixels, with a single pixel real-space size of 11.4 nm. Also shown is a resolution threshold curve based on a half-bit of information per pixel. The FRC resolutions given in the text refer to when each curve first crosses the threshold.

**Table 1.** FRC-derived resolution for images acquired under differing optical aberrations. SRRF data from figure 8, TPE data derived from FRC of the original image stacks used to generate the SRRF images.

| aberration | SRRF FRC (Pixels$^{-1}$) | SRRF FRC (nm) | TPE FRC (nm) |
|---|---|---|---|
| none | 0.269 | 42.5 | 250 |
| low | 0.235 | 49 | 262 |
| high | 0.062 | 184 | 304 |

frequency (minimum feature size) that still contains information in the images. Figure 8 shows the FRC curves derived from each aberration case, with the FRC value plotted against inverse pixels as the unit of spatial frequency. To generate the FRC curve, we performed drift correction on the original TPE image sequences, deinterleaved the resulting odd and even frames and processed them in the SRRF algorithm. The resulting pair of SRRF images was then used for the FRC algorithm. Our chosen SRRF parameters resulted in a pixel size of 11.4 nm, which can be used to convert from spatial frequency to resolution.

Also shown in figure 8 is the threshold curve used to determine the FRC resolution. Common approaches to determine the FRC resolution use a fixed value of 1/7 as the threshold for the FRC resolution: the spatial frequency at which the FRC curve first falls under 1/7 is given as the resolution. A problem with this is, for low spatial frequencies, the FRC curve is derived from Fourier-domain rings that contain fewer pixels than those for higher spatial frequencies. There is therefore a difference in the statistical behaviour in these two conditions [49,50] for which a better threshold is the curve defining the FRC level at which each pixel contains half of one bit of information. We include this 'half-bit threshold' in figure 8 and use it to determine the resolution for each of our images. The resulting resolutions are given in table 1.

That the FRC resolution agrees with the FWHM of the Gaussian fit in figure 6b is gratifying. We note that the FRC measures self-similarity of images, and does so in a manner that averages across the entire image. As such, the number obtained from the FRC curve, and which is generally considered to be equivalent to the resolution, does not necessarily hold across the entire image. Indeed, despite the FRC resolution of 49 nm, the upper central pair of ND in figure 7d are not resolved. We therefore emphasize again that the FRC is a useful tool, but that there is still a widely acknowledged, and unsolved, problem of the definition of image resolution in super-resolution microscopy. What is

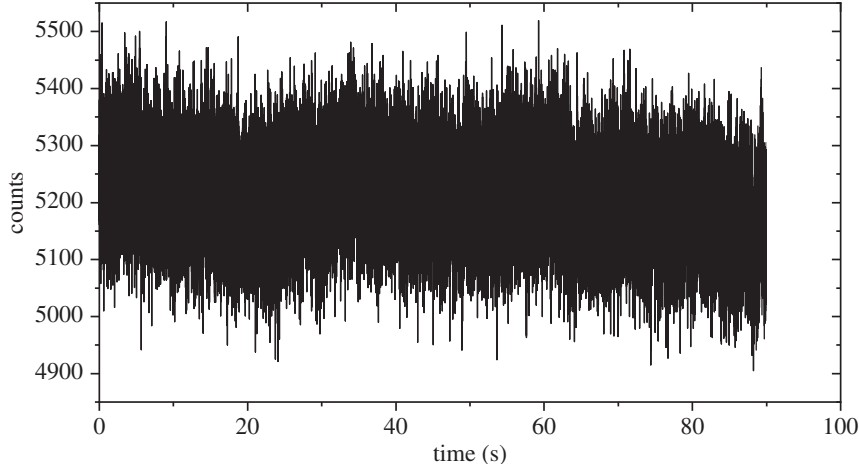

**Figure 9.** Demonstration of the photostability of ND under continuous single-photon excitation. Laser power was 1 mW at 532 nm using 40 MHz repetition rate. Sample points were generated consecutively with a 2 ms detector integration time.

nevertheless surprising is the robustness of the SRRF algorithm to aberrations—we see evidence of resolution enhancement even in the case of strongly aberrated images, although it is doubtful how much information can usefully be extracted from the SRRF image in the high aberration case.

## 6. Nanodiamond photostability

To further demonstrate the exceptional photostability of ND, we prepared a sample of 70 nm ND (Sigma Aldrich) on a #1.5 coverslip using immersion oil as the mounting medium. Under 532 nm excitation, we were able to image a single ND over multiple hours without any loss of signal. Figure 9 shows an example of this photostability; there is little change in the fluorescence detected over a period of 100 s of continuous excitation of a single ND using 1 mW of average power at 532 nm. Each data point was taken sequentially with a 2 ms integration time: this plot shows $5 \times 10^4$ fluorescence measurements. The small loss of signal by the end of the run is entirely due to sample drift and can be fully recovered by realigning the excitation laser to the ND location. Under TPE conditions, we observed directly equivalent stability. By way of comparison with conventional fluorophores, with our microscope we typically observe a permanent decrease in fluorescence of 50% in samples labelled with Alexa 488 after 100 frames of 1.5 ms pixel duration and 100 µW of excitation power. With twice-Nyquist sampling conditions, this implies an effective photobleaching half-life of 2.4 s. Thus, after an equivalent 100 s duration experiment to that shown in figure 9, the Alexa 488 would be emitting at an intensity $3 \times 10^{-13}$ of which it began.

## 7. Conclusion

ND has been shown to have much promise as a fluorophore for use in biological imaging. With this study, we have shown that it is not only an exceptional fluorophore in single-photon imaging, but that it is also superb when used with two-photon excitation microscopes. Furthermore, we demonstrate that its photostability gives it significant advantages when performing adaptive-optics or computational super-resolution images. Indeed, we saw evidence for a ten-fold resolution increase when generating SRRF images from our two-photon data.

Data accessibility. All data underpinning this publication, along with descriptions of the file format generated by the custom software running our microscope and scripts for importing the data into Fiji, are openly available from the University of Strathclyde KnowledgeBase at https://doi.org/10.15129/2365ed7a-4695-4928-b748-44308585b1a2.
Competing interests. We declare we have no competing interests.
Authors' contributions. G.E.J. constructed the microscope, participated in the data acquisition, performed data analysis and drafted the manuscript; G.S.C. constructed the microscope, performed data acquisition and microscope calibration experiments; B.R.P. conceived of the work, constructed the microscope, participated in the data acquisition, performed data analysis and helped draft the manuscript. All authors gave final approval for publication.

Funding. This work was funded under grants from the Royal Society (RGF\EA\181058 and URF\R\180017) and EPSRC (EP/M003701/1). G.S.C. is funded under 'OPTIMA: The EPSRC and MRC Centre for Doctoral Training in Optical Medical Imaging'. B.R.P. holds a Royal Society University Research Fellowship.

Acknowledgements. We wish to acknowledge the loan of a Fidelity 2 laser from Coherent Inc., which was used to generate all two-photon excitation data in this paper.

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
