## [Reviewer comments · Royal Society Open Science]

Review History

RSOS-190589.R0 (Original submission)

Review form: Reviewer 1

Is the manuscript scientifically sound in its present form?

Yes

Are the interpretations and conclusions justified by the results?

Yes

Is the language acceptable?

Yes

Is it clear how to access all supporting data?

No

Do you have any ethical concerns with this paper?

No

Have you any concerns about statistical analyses in this paper?

No

Recommendation?

Accept with minor revision (please list in comments)

Comments to the Author(s)

This manuscript outlines the exploitation of two-photon excitation (TPE) and adaptive optics (AO) for imaging fluorescent nanodiamonds (NDs). It presents, in relatively clear terms (and language that is accessible), the advantage of combining these three components to well-surpass the diffraction limit. By adopting the SRRF approach to image analysis, the authors emphasise the importance of proper aberration correction for achieving the promised super resolution. I would like the authors to address the following points.

Minor concerns:

1. Line 7 of page 3 outlines the benefits of AO by citing 3 papers. In comparison to the benefits of TPE outlined in the previous page, this section does not adequately introduce the principles or the applications of AO, particularly to the naïve reader.
2. Line 22 of page 7: What are the specifications of the coverslips used?
3. Line 42 of page 7: The authors equate the optical fibre in the emission light path to an effective pinhole. What is the effective pinhole size? Does this do anything to improve the contrast of the recorded image?
4. The description of the optical path and the deformable mirrors have been described briefly in the text. A schematic diagram (in the form of a supplement) would be useful for a non-specialist reader.
5. Looking at the raw data for Figure 1, it looks as though the structures in the TPE images are better-shaped (more compact and less ringing) compared to SPE. Why is this?

Major concerns:

1. I appreciate that the authors use the in-built drift correction function in NanoJ. For the benefit of those who are not familiar with NanoJ, it is perhaps important to outline how this correction works. The raw data that have been shared appear to drift by 90+ nm. What is the limit of drift tolerance with this approach?
2. The Fourier ring correlation (FRC) has been used to characterise the improvement in the resolution gained through the aberration correction in the primary data used for SRRF. FRC, however, is also very sensitive to drift. Were these datasets corrected for drift before splitting into two independent time-stacks for SRRF?

General considerations:

1. The authors outline the potential issues of photobleaching in using (conventional) fluorophores in combination with TPE and AO. Demonstrating this point through a side-by-side comparison with a bleach-susceptible marker (e.g. fluorescent latex microspheres) would have added value in demonstrating this point, particularly to readers who are not familiar with NDs. The lack of this comparison however does not diminish the validity of the point made in this manuscript.
2. The authors conclude that this methodology offers strong promise for super-resolution imaging biological samples. It would have been more confidence inspiring to see an application of this. From the text of the paper, I gather this to be a future development planned by the authors.

Review form: Reviewer 2**Is the manuscript scientifically sound in its present form?**

Yes

Are the interpretations and conclusions justified by the results?

Yes

Is the language acceptable?

Yes

Is it clear how to access all supporting data?

Yes

Do you have any ethical concerns with this paper?

No

Have you any concerns about statistical analyses in this paper?

No

Recommendation?

Accept with minor revision (please list in comments)

Comments to the Author(s)

The manuscript by Johnstone, Cairns and Patton describes the use of nanodiamond as a fluorophore in 2photon imaging, also demonstrating super-resolution imaging via the SRRF technique. The manuscript is well written and referenced, and contains interesting experimental findings. There is a high level of consideration given to understanding the results, which is refreshing. The work is well suited to RS Open Science and I would recommend it for publication. I have several points that the authors may wish to comment on:

1. In the conclusion, it states "Furthermore, we demonstrate that its photostability gives it significant advantages when performing adaptive-optics or computational super-resolution images." In fact, I couldn't see any data in the manuscript on photostability, which should really be added, rather than just asserted.
2. Page 8, line 51. The authors say that they correct for in-plane sample drift, they should give what this value typically is- this will give some indication of the z drift present, which will probably be at least as bad. On an associated theme, it would be useful to state the acquisition time for 100 frames.
3. Figure 4 shows a 2d slice the two photon PSF. While in this system, it does correspond to the TPE PSF, this could be slightly misleading as this microscope is highly unusual for a two photon system in that it has a confocal pinhole for the detection. This is particularly the case when demonstrating the adaptive optics addition of aberration, since the DM adds the astigmatism in the detection path also.
4. The 4x enhancement of the emission in the two-photon excitation is very interesting. Is it definitely the same isolated ND being imaged? While the rep. rate can be used to explain some of this, there is still a big enhancement and no sign of saturation. Why would there be no NV0 emission on the 1pE, but it becomes apparent in 2PE? Are the authors sure the 2PE signal is indeed NV or could there be something else here?
5. Figure 7 - perhaps it is also useful here to provide count rates. How does this correspond to the estimated Strehl given the wavefront error? Page 10, line 45, is the sigma measured?
6. I found the FRC discussion very interesting. However, I am unsure how reliable the FRC might be as a characterisation tool? The authors state that the FRC may be used to better characterise the effective resolution. For Strehl ratio of 0.3, the result is given as 49 nm, but in Fig 7d, it seems we are no longer able to resolve the pair of NDs at the top of the image? The FRC for Strehl of 0.1 is 184nm, still very respectable in terms of resolution, but the SRRF image appears as noise with many artefacts, and I am not sure it really offers any improvement over the raw image?

Decision letter (RSOS-190589.R0)

18-Jun-2019

Dear Dr Patton

On behalf of the Editors, I am pleased to inform you that your Manuscript RSOS-190589 entitled "Nanodiamonds enable adaptive-optics enhanced, super-resolution, two-photon excitation microscopy" has been accepted for publication in Royal Society Open Science subject to minor revision in accordance with the referee suggestions. Please find the referees' comments at the end of this email.

The reviewers and handling editors have recommended publication, but also suggest some minor revisions to your manuscript. Therefore, I invite you to respond to the comments and revise your manuscript.

- Ethics statement

- Data accessibility

<http://datadryad.org/submit?journalID=RSOS&manu=RSOS-190589>

- Competing interests

- Authors' contributions

- Acknowledgements

- Funding statement

Because the schedule for publication is very tight, it is a condition of publication that you submit the revised version of your manuscript before 27-Jun-2019. Please note that the revision deadline will expire at 00.00am on this date. If you do not think you will be able to meet this date please let me know immediately.

- 1) A text file of the manuscript (tex, txt, rtf, docx or doc), references, tables (including captions) and figure captions. Do not upload a PDF as your "Main Document";
- 2) A separate electronic file of each figure (EPS or print-quality PDF preferred (either format should be produced directly from original creation package), or original software format);
- 3) Included a 100 word media summary of your paper when requested at submission. Please ensure you have entered correct contact details (email, institution and telephone) in your user account;
- 4) Included the raw data to support the claims made in your paper. You can either include your data as electronic supplementary material or upload to a repository and include the relevant doi

within your manuscript. Make sure it is clear in your data accessibility statement how the data can be accessed;

5) All supplementary materials accompanying an accepted article will be treated as in their final form. Note that the Royal Society will neither edit nor typeset supplementary material and it will be hosted as provided. Please ensure that the supplementary material includes the paper details where possible (authors, article title, journal name).

If your manuscript is newly submitted and subsequently accepted for publication, you will be asked to pay the article processing charge, unless you request a waiver and this is approved by Royal Society Publishing. You can find out more about the charges at <http://rsos.royalsocietypublishing.org/page/charges>. Should you have any queries, please contact opscience@royalsociety.org.

Kind regards,
Alice Power
Editorial Coordinator
Royal Society Open Science
opscience@royalsociety.org

on behalf of Dr Tufarelli Tommaso (Associate Editor) and Miles Padgett (Subject Editor)
opscience@royalsociety.org

Associate Editor Comments to Author (Dr Tufarelli Tommaso):

Please reply to all the points raised by the referees in your resubmission, and where applicable explain how you have amended the manuscript to take them into account.

Reviewer comments to Author:
Reviewer: 1

Comments to the Author(s)

This manuscript outlines the exploitation of two-photon excitation (TPE) and adaptive optics (AO) for imaging fluorescent nanodiamonds (NDs). It presents, in relatively clear terms (and language that is accessible), the advantage of combining these three components to well-surpass

the diffraction limit. By adopting the SRRF approach to image analysis, the authors emphasise the importance of proper aberration correction for achieving the promised super resolution. I would like the authors to address the following points.

Minor concerns:

1. Line 7 of page 3 outlines the benefits of AO by citing 3 papers. In comparison to the benefits of TPE outlined in the previous page, this section does not adequately introduce the principles or the applications of AO, particularly to the naïve reader.
2. Line 22 of page 7: What are the specifications of the coverslips used?
3. Line 42 of page 7: The authors equate the optical fibre in the emission light path to an effective pinhole. What is the effective pinhole size? Does this do anything to improve the contrast of the recorded image?
4. The description of the optical path and the deformable mirrors have been described briefly in the text. A schematic diagram (in the form of a supplement) would be useful for a non-specialist reader.
5. Looking at the raw data for Figure 1, it looks as though the structures in the TPE images are better-shaped (more compact and less ringing) compared to SPE. Why is this?

Major concerns:

1. I appreciate that the authors use the in-built drift correction function in NanoJ. For the benefit of those who are not familiar with NanoJ, it is perhaps important to outline how this correction works. The raw data that have been shared appear to drift by 90+ nm. What is the limit of drift tolerance with this approach?
2. The Fourier ring correlation (FRC) has been used to characterise the improvement in the resolution gained through the aberration correction in the primary data used for SRRF. FRC, however, is also very sensitive to drift. Were these datasets corrected for drift before splitting into two independent time-stacks for SRRF?

General considerations:

1. The authors outline the potential issues of photobleaching in using (conventional) fluorophores in combination with TPE and AO. Demonstrating this point through a side-by-side comparison with a bleach-susceptible marker (e.g. fluorescent latex microspheres) would have added value in demonstrating this point, particularly to readers who are not familiar with NDs. The lack of this comparison however does not diminish the validity of the point made in this manuscript.
2. The authors conclude that this methodology offers strong promise for super-resolution imaging biological samples. It would have been more confidence inspiring to see an application of this. From the text of the paper, I gather this to be a future development planned by the authors.

Reviewer: 2

Comments to the Author(s)

The manuscript by Johnstone, Cairns and Patton describes the use of nanodiamond as a fluorophore in 2photon imaging, also demonstrating super-resolution imaging via the SRRF technique. The manuscript is well written and referenced, and contains interesting experimental findings. There is a high level of consideration given to understanding the results, which is refreshing. The work is well suited to RS Open Science and I would recommend it for publication. I have several points that the authors may wish to comment on:

1. In the conclusion, it states "Furthermore, we demonstrate that its photostability gives it significant advantages when performing adaptive-optics or computational super-resolution images." In fact, I couldn't see any data in the manuscript on photostability, which should really be added, rather than just asserted.
2. Page 8, line 51. The authors say that they correct for in-plane sample drift, they should give what this value typically is- this will give some indication of the z drift present, which will

probably be at least as bad. On an associated theme, it would be useful to state the acquisition time for 100 frames.

3. Figure 4 shows a 2d slice the two photon PSF. While in this system, it does correspond to the TPE PSF, this could be slightly misleading as this microscope is highly unusual for a two photon system in that it has a confocal pinhole for the detection. This is particularly the case when demonstrating the adaptive optics addition of aberration, since the DM adds the astigmatism in the detection path also.

4. The 4x enhancement of the emission in the two-photon excitation is very interesting. Is it definitely the same isolated ND being imaged? While the rep. rate can be used to explain some of this, there is still a big enhancement and no sign of saturation. Why would there be no NV0 emission on the 1pE, but it becomes apparent in 2PE? Are the authors sure the 2PE signal is indeed NV or could there be something else here?

5. Figure 7 - perhaps it is also useful here to provide count rates. How does this correspond to the estimated Strehl given the wavefront error? Page 10, line 45, is the sigma measured?

6. I found the FRC discussion very interesting. However, I am unsure how reliable the FRC might be as a characterisation tool? The authors state that the FRC may be used to better characterise the effective resolution. For Strehl ratio of 0.3, the result is given as 49 nm, but in Fig 7d, it seems we are no longer able to resolve the pair of NDs at the top of the image? The FRC for Strehl of 0.1 is 184nm, still very respectable in terms of resolution, but the SRRF image appears as noise with many artefacts, and I am not sure it really offers any improvement over the raw image?

Author's Response to Decision Letter for (RSOS-190589.R0)

See Appendix A.

Decision letter (RSOS-190589.R1)

04-Jul-2019

Dear Dr Patton,

I am pleased to inform you that your manuscript entitled "Nanodiamonds enable adaptive-optics enhanced, super-resolution, two-photon excitation microscopy" is now accepted for publication in Royal Society Open Science.

on behalf of Dr Tufarelli Tommaso (Associate Editor) and Miles Padgett (Subject Editor)
openscience@royalsociety.org

Appendix A

rsos.royalsocietypublishing.org

Research

Article submitted to journal

Subject Areas:

Physics, Microscopy, Materials
Science

Keywords:

Nanodiamond , Super-resolution ,
Multi-photon Excitation , Adaptive
Optics

Author for correspondence:

Brian Patton

e-mail: brian.patton@strath.ac.uk

Nanodiamonds enable adaptive-optics enhanced, super-resolution, two-photon excitation microscopy

Graeme E. Johnstone¹, Gemma S.
Cairns¹, and B. R. Patton¹

¹Department of Physics and SUPA, University of
Strathclyde, Glasgow G4 0NG, Scotland, United
Kingdom

This manuscript version highlights our response to reviewers' comments. Changes to text are *emphasised in bold, italic font*

Introduction

The biocompatibility of diamond [1], combined with the ability to make nanoscopic particles of <100 nm diameter, has led to research into the use of nanodiamond (ND) for a variety of biological applications [2], including drug delivery [3,4] and use as a fluorescent marker for microscopy [5,6]. The surface chemistry of ND allows it to be functionalised by chemically attaching a range of different molecules to the ND surface [7,8]. Examples of functionalisation strategies include antibody [9,10] and DNA [11] binding. The benefits arise by allowing efficient targetting of ND to subcellular features of interest, thereby allowing effective labelling for fluorescent microscopy [2].

The fluorescence of ND is due to defects in the diamond structure [12]. A large number of defects are known to show fluorescent properties [8], but of particular interest is the nitrogen-vacancy (NV) defect, which occurs when two neighbouring carbon atoms are replaced by a nitrogen atom and a vacant space in the crystal lattice. This defect is most commonly found in a negatively charged state, referred to as NV⁻, and is efficiently imaged with a single photon excitation of 532 nm that results in emission over a wide wavelength band from 637 nm to approximately 800 nm. A particular advantage of NV⁻ emission is that, unlike most conventional fluorophores used for microscopy, it is completely photostable, allowing imaging of individual cells labelled with ND over periods of a week or more [13]. The brightness of an individual ND is a function of the number of defects it contains. This number can be increased during the production of nanodiamond, allowing for even brighter labelling, and is typically correlated with the size of the ND: particles smaller than 5 - 10 nm are unlikely to be able to support a fluorescently active NV [14,15] without specific processing [16]. By contrast, 100 nm particles may have hundreds of active NV centres. This flexibility allows a tradeoff between the intrinsic brightness of the ND (improving signal to noise for a given image acquisition rate) and the potential impact of the ND on the local biological processes (due either to physical blocking or surface chemistry effects). There have been many studies demonstrating the use of ND with standard microscopy techniques, including confocal microscopy [17] and super-resolution imaging in the form of stimulated emission depletion microscopy [18,19].

An additional driver for the interest in NV as a fluorophore relates to the quantum mechanical properties that can be accessed optically [20]. By combining optical excitation and readout (as is performed when imaging in a microscope) with some pulsed microwave (≈ 2.88 GHz) electronic-spin manipulation, it is possible to use the NV⁻ centre as a sensor that can detect a number of features of the local environment, such as the temperature [6,21] and the presence of magnetic fields [22–25]. Active sensing of environmental changes induced by biological processes at a sub-micron scale, with NV centres, would extend the usefulness of ND in a range of imaging applications. However, while the emission of NV⁻ is in a wavelength range with low scattering and absorption by water, the single photon excitation wavelength of 532 nm poses problems related both to scattering and autofluorescence effects that will decrease the effective sensitivity of the ND to the signals of interest.

One alternative approach is to take advantage of two-photon excitation (TPE) microscopy, which has the ability to image at a greater depth than single photon microscopy [26] due to decreased scattering and absorption. There is also a better signal to noise ratio as the fluorescence is only generated at the excitation focus and not throughout the whole illumination cone, giving significant benefits in autofluorescent samples. It has recently been shown that NV centres in bulk diamond samples can fluoresce with TPE [27] and likewise for NV in fragments of diamond of sizes 10 - 100 μm [28]. Tuning the excitation wavelengths in a range from 1030 to 1310 nm allows preferential excitation of NV⁻ or the neutral NV⁰ state. There have been initial reports of TPE fluorescence in nanodiamonds [29–31], showing that they are compatible with this mode of imaging.

In this paper we further demonstrate TPE imaging of NV centres in ND and show how ND is particularly suited to computational super-resolution techniques, such as that enabled by the super-resolution radial fluctuations (SRRF) [32] approach. *To allow high quality super-resolution imaging, we take advantage of the adaptive optics technology incorporated in our microscope. By using deformable mirrors, which can introduce controllable distortions to the wavefront of the light that propagates through the microscope, it is possible to compensate for the optical aberrations introduced by the inhomogenous nature of the samples we wish to image. For further understanding of the requirements for implementation of adaptive optics within microscopy, we recommend Refs. [33,34].* While multi-photon microscopy is inherently suited to imaging in optically aberrating samples, it nevertheless benefits from adaptive optical image correction [35–37]. With this in mind, we also demonstrate adaptive optic control of the excitation focal volume and the resulting changes in the images obtained from single ND crystals. By combining adaptive-optics and computational super-resolution imaging, we also show that ND is a superb fluorophore with significant potential for applications in which efficient correction of aberrations deep within an aberrating medium (such as tissue) is essential.

Experimental

Materials and sample preparation

The nanodiamonds used in these experiments were produced by Adamas Nanotechnologies. We prepared slides for imaging from a mix of two monodisperse suspensions (both 0.1%w/v) of 40 nm and 100 nm diameter ND. The 40 nm ND each contain approximately 10 NV while there are closer to 400 NV per 100 nm ND as per the manufacturers calibration information. To prepare a suspension suitable for deposition on a coverslip we first sonicated each source of NDs to break up larger aggregates before adding 10 μ l of each ND suspension to 100 μ l of distilled water. The resulting suspension was then deposited onto a #1.5 microscope glass cover slip and allowed to dry to ensure some ND adhered to the coverslip before being mounted on to a microscope slide using a small amount of distilled water as a mountant medium and finally sealing the sample with nail polish.

Microscope and imaging equipment

The microscope that provided all of the imaging for this work is a custom-designed, confocal microscope. A 1.35NA oil immersion objective lens was used for imaging in an epi-flourescence geometry. We used two excitation lasers for the work presented in this paper:

- A <50 ps pulse-duration 532 nm Picoquant laser running at a repetition rate of 40 MHz. This laser was used for single photon excitation of NV- centres.
- A Coherent Fidelity 2 pulsed fibre laser with a wavelength centred on 1070 nm, a bandwidth of 70 nm and a pulse duration of 40 fs at the laser's output. A chirp compensator allows optimisation of the pulse duration at the sample by maximising the observed multi-photon signal.

The fluorescent light emitted from the sample was coupled via an optical fibre (acting as the confocal pin hole for the single photon excitation) to a single pixel detector, a Laser Components avalanche photodiode (Count-50). *The final lens before the optical fibre was chosen to set an effective pinhole size of 1 Airy unit. This may give a slight increase in contrast in two-photon imaging mode, however the effect of the pinhole will be dominated by the inherent optical sectioning provided by the TPE process.* Images were created by raster scanning the beam across the sample using a Newport Fast Steering Mirror (FSM-300) and axial-imaging was performed by moving the objective which was mounted on a Physik Instrumente piezo positioner (P-725 PIFOC). We have chosen the detection path to have a wavelength sensitivity of 650-750 nm.

This range corresponds with the peak of the emission from NV- centres. As will be discussed further in section 4, a Boston Micromachines Corporation deformable mirror (Multi-DM) with 140 actuators and $3.5\mu\text{m}$ stroke is used for aberration correction of both the excitation and signal paths. Control of the microscope was performed by custom-written Labview software driving a National Instruments FPGA.

Two Photon Imaging

We begin by demonstrating the effectiveness of two-photon imaging with our system. Figure 1 shows a comparison of one-photon and two-photon imaging on our mixed ND sample. We processed the data initially in Fiji [38,39] and then generated the output figures for publication in OriginPro. We present the images using “CubeHelix” [40], a colourmap that increases linearly in perceived brightness, thereby making it suitable for colour-blind readers and reproduction in greyscale media. All unprocessed imaging data is available at the Strathclyde file sharing site along with descriptions of the file format generated by the custom software running our microscope and scripts for importing the data into Fiji.

Figure 1 shows the comparison between single-photon and multi-photon imaging of a single region of our sample. Sub-figures 1 a) and c) show that there is excellent correlation between the ND's that appear when imaging with 532 nm excitation and those present with 1070 nm excitation, respectively. To show the comparable resolution in both cases, Figs. 1 b) and d) show a higher resolution scan of the same pair of ND within the larger scan.

An advantage of multi-photon excitation is that it provides intrinsic sectioning when imaging, due to the excitation only occurring in the central volume of the excitation point-spread function (PSF). We include a video, (Supplemental Material 1), which shows multiple ND at different z-positions within the sample when imaged with the 1070 nm excitation.

While Fig. 1 demonstrates that we are seeing multi-photon excitation of the NV within our ND, we also wanted to confirm that it is a two-photon process. To confirm this, the beam was centred on a single ND and the detected photon counts were measured versus excitation power for both single- and multi-photon excitation modes. The results are shown in figure 2 for the 532 nm excitation and in figure 3 for the 1070 nm excitation. In the 532 nm excitation case, and at low input powers, the number of counts increases linearly with the power of the excitation laser. This is as expected for single photon fluorescence. For all fluorophores, as the excitation power increases further, the emitted fluorescence increases sub-linearly and starts to plateau - it saturates [41]. Figure 2 shows exactly this behaviour in the single-photon excitation of ND, with saturation already apparent at $100\mu\text{W}$ incident power and an estimated peak count rate of approximately $80,000\text{ s}^{-1}$. The inset of Fig. 2, plotting count rate and power on logarithmic scales, shows a power law fit with power parameter $n = 0.8$ at low excitation power. That the value of n is less than one likely reflects the fact that we included points where saturation is already beginning to occur. It is worth noting here that it is often undesirable to image fluorophores in the saturation regime as it increases the probability of a photo-bleaching event due to inter-system crossings when the fluorophore is in the excited state. However, the photostability of NV centres means that it is possible to image them indefinitely even when well into the saturation regime. This is particularly relevant when seeking to use the emission properties of the NV as an environmental sensor. Knowing the relationship between excitation power and degree of saturation is useful as it allows us to maximise the photons being detected while decreasing the possibility of other phototoxicity effects from the excitation laser.

In the case of the 1070 nm excitation, Fig. 3, we see a super-linear dependence of the emission on the excitation power. For a pure two photon process the count rate should be quadratic with the excitation intensity. Performing a power law fit we recover a power parameter of $n = 1.84 \pm 0.02$ indicating that we are seeing TPE as the mechanism for generating fluorescence at this excitation wavelength. There are two phenomena here that are worth commenting on. Firstly, we do not see saturation in the TPE power dependence. As discussed in the description of the microscope, we are limited to 12 mW power at the sample for the 1070 nm excitation laser, so this is likely a result

Figure 1. a) Cluster of nanodiamonds imaged with single-photon excitation confocal microscopy using 532 nm laser excitation. b) A higher resolution 532 nm excitation scan of the region highlighted in a). c) The same region as a), imaged in two-photon mode with 1070 nm excitation. d) A higher resolution 1070 nm excitation scan of the region highlighted in c). The sample contains a mixture of 40 nm and 100 nm diameter nanodiamonds. The colour scale for all images is linear and normalised to the maximum and minimum pixel values of each individual image.

of insufficient available laser power. The second point of note is that the TPE appears to show higher fluorescence counts than the single-photon excitation at saturation. We do not currently have a full explanation for this. One contribution to the observed count rates comes from the fact that the 1070 nm laser was operated at a higher pulse repetition rate than the 532 nm laser (70 MHz vs 40 MHz). For short-duration laser pulses ($\tau_{Pulse} < \tau_{NV-}$) this is equivalent to stating that each pulse results in one photon being emitted. Therefore, for repetition rates $< 1/\tau_{NV-}$, as is the case here, we would expect the count rate to be proportional to the laser repetition rate. Reference [28] also shows that, even with the 650 nm long pass filter in our system, there can still be significant NV0 emission leaking into the NV- channel. Furthermore, with 1070 nm excitation there is a significant cross-section for NV0 excitation. We did not have appropriate spectroscopic equipment to unambiguously determine the contribution of the NV0 to this excess count rate. However, we note that this would mainly be an issue for the use of the NV- spin properties in

Figure 2. Observed fluorescence count rate versus excitation power for single-photon excitation with 532 nm laser. Inset: Logarithmic power dependence showing sub-linear power law behaviour even at low excitation intensities. Scale is $1 \text{ dBm} = 1 \text{ mW}$ as measured at sample

local field sensing. For general two-photon imaging, the apparent brightness of the ND with the excitation wavelength and power demonstrated here is beneficial. Should the application require efficient NV- only excitation, then reference [28] suggests longer wavelength excitation ($>1150 \text{ nm}$) will improve the emission from NV-.

Adaptive Optics

Typical implementations of adaptive optics in two-photon microscopy [35] use active elements in the excitation path only and, due to the signal being generated only at the focal volume, employ wavefront sensors to directly detect the aberrations. As we are using a confocal detection scheme, the microscope used for these measurements has a deformable mirror (DM) installed in the common beam path and in such a way that it is imaged onto the back aperture of the objective. Deformations of the mirror can therefore be used to implement adaptive optical control of the PSF, with particular emphasis on correcting aberrations induced by the sample and which degrade image quality. Orthogonal control modes for the DM are derived from the individual actuator influence functions as described in references [42,43] and we use a sensorless correction procedure [36,44] in which image-based metrics are used to measure the optimal shape of the DM. A driver for the development of efficient AO correction schemes has been the desire to minimise the photons lost to the correction process; if the sample undergoes significant photobleaching during the aberration estimation then it may call into question the use of AO in that particular sample. The lack of photobleaching of the NV centre therefore makes it an ideal candidate for implementing and testing aberration correction techniques.

Figure 3. Observed fluorescence count rate versus excitation power for multi-photon excitation with 1070 nm laser. Inset: Logarithmic power dependence showing approximately quadratic power law behaviour (exponent $n = 1.8$) indicating two-photon excitation. Scale is $1 \text{ dBm} = 1 \text{ mW}$ as measured at sample

In figure 4 we demonstrate control over the PSF of our two-photon confocal imaging system. All sub-figures are of the same ND, differing in the aberration applied to the DM and with a colour scale normalised to the maximum pixel value shown in the sub figure. Since aberrations are expected to decrease the peak intensity of the PSF, we also show this peak value as referenced to the peak intensity of the unaberrated PSF, Fig. 4a). For this demonstration we have applied modes corresponding to the two primary astigmatism aberrations (Zernike modes 5 and 6, using Noll indices) . Fig. 4b) shows vertical astigmatism, while Fig. 4c) and d) show positive and negative amounts of oblique astigmatism. The fidelity with the expected PSF shape is good, with little cross-talk between modes. The combination of two photon excitation microscopy, ND as a fluorophore and adaptive optics will allow deeper imaging in tissue than possible with either technique alone and we look forward to further exploring the possibilities in future work.

Superresolution imaging of two photon excitation microscopy

The photostability of the NV defect makes it particularly suited to computational super-resolution techniques such as Super-Resolution Radial Fluctuations (SRRF) imaging. The algorithm and theory underlying the method is described in Reference [32], and a plug-in module for ImageJ makes it a particularly user-friendly technique. SRRF makes use of the radial symmetry of the PSF to generate a radially map that locates the fluorophore within the image. By then taking advantage of correlations in naturally occurring fluctuations in the radial symmetry measured over a large number of images of the same object, the SRRF algorithm can extract super-resolution images. In practice, a minimum of 100 standard resolution images are synthesised into a single

Figure 4. Experimental demonstration of deformable-mirror based control of the two-photon point-spread function (PSF). a) Optimally corrected PSF. Aberrated with: b) Vertical astigmatism c) Positive oblique astigmatism d) Negative oblique astigmatism. All scale bars are 500 nm, the colourscale is linear and ranges down from the given maximum pixel value (shown relative to the peak of a), the brightest image) to 10% of that value for each sub-figure.

SRRF image. It is worth noting that current versions of the SRRF algorithm do not offer any resolution enhancement along the optical axis.

Figure 5 a) shows a pair of ND clusters imaged using two photon excitation microscopy. This image was generated by repeating 100 times and processed by correcting for in-plane sample drift, using the NanoJ-Core algorithm [45], followed by summing the 100 frames. *With a 1.5 ms pixel dwell time, and an image size of 101x101 pixels, each frame takes approximately 18 s to complete (including line flyback time), with the full image stack therefore taking approximately 30 minutes.* To generate the SRRF image shown in Fig. 5 b) we again applied the drift-correction routine before using the NanoJ-SRRF plugin with settings: Ring Radius = 0.5, Radiality Magnification = 6, Axes in Ring = 7 and Temporal Radiality Average. The resultant image shows significant increase in resolution, however there are some artefacts present; along with the lines joining individual NDs, inspection of the raw data shows predominant sample drift along the Z axis and the presence of two distinct layers of ND separated in the Z-axis. *From*

Figure 5. Super-Resolution Radial Fluctuations (SRRF) imaging of two-photon excitation ND samples a) Drift-corrected sum of 100 acquisitions in TPE acquisition mode b) Super-resolution image derived from original image stack. Scale bar 1 μm , linear colour scale applied to each image.

the drift correction table generated by NanoJ, which corrects only for lateral drift, we see this corroborated with a maximum lateral drift of 45 nm. Supplemental Movie 1 shows the results of a 3D stack of this cluster.

To better demonstrate the quality of SRRF imaging possible with ND, we performed further measurements on the upper cluster of nanodiamond imaged in Fig. 5. By keeping the pixel size approximately the same (70 nm versus 80 nm in Fig. 5) but decreasing the image boundaries, we hoped that the resulting shorter scan times would decrease the sample drift along the optical axis. Drift in this axis cannot be efficiently corrected on our hardware and has a significant deleterious effect on the SRRF processing algorithm. The results of these measurements are shown in Figure 6, with the 100 drift-corrected and summed standard resolution TPE frames shown in Fig. 6 a) and the SRRF processed image in Fig. 6 b). *In this case the frame time is about 5 s and while the maximal lateral drift as determined by the drift correction routine is now 63 nm, there is much less Z-axis drift apparent in the raw data.* As is the case with the larger clusters in Fig. 5, the SRRF algorithm is able to recover significant extra resolution, and the higher quality initial imaging translates into higher SRRF resolutions. We will return to this point below. To show the enhancement in resolution, Fig. 6 b) includes insets that enlarge two 250 nm square regions of interest. In the lower region we show that there is structure now visible within the brightest diffraction-limited spot within the TPE image of Fig. 6 a). The lateral extent of this emitting region is of the order of 100 nm, however we do not have sufficient information to know unambiguously whether we are observing structured emission from a single 100 nm ND or emission from a small cluster of 40 nm ND. The upper region highlighted on Fig. 6 b) appears to be clearer in interpretation; there are two closely spaced emitters that the SRRF map shows as being distinguishable. To gain an initial estimation of the resolution of the SRRF image, we show a vertical slice through the left emitter of the pair along the green dashed line. Fitting this intensity profile to a Gaussian gives a 44 nm full-width-half-maximum (FWHM), consistent with this being a single 40 nm ND.

We have shown that sample drift can adversely affect the quality of a SRRF reconstruction. An advantage of our microscope, when combined with the lack of photobleaching of the ND, is that we can also explore the effects of optical aberrations on the SRRF process. Figure 7 shows a demonstration of how this can be put into practice. As a reference, the upper row of Fig. 7 (a) and b)) shows the unaberrated TPE and SRRF images. We then acquired further TPE image

Figure 6. Higher frame-rate imaging of upper cluster of ND from Fig. 5 a) Drift-corrected sum of 100 acquisitions in TPE acquisition mode b) Super-resolution image derived from original image stack. Insets are from two, 250 nm square regions as highlighted on main SRRF image. Upper inset also shows intensity plot along a vertical slice through one of the SRRF-resolved emitters. Indicated is the 44 nm full-width-half-maximum of a Gaussian fit to this intensity profile. 1 μm scale bars refer to main images, linear colour scale applied to each image.

sets with identical imaging conditions apart from a known aberration being applied to the DM. For the middle row (Fig. 7 c) and d)) we applied 1 unit each of astigmatism and coma (Zernike Noll indices 6 and 7). To further quantify the effect of the aberration, we consider the effective RMS wavefront error, σ , being applied by the DM. In this case $\sigma = 1.10$ rad. *From this, we can also estimate the Strehl ratio as $S = e^{-\sigma^2}$. The Strehl ratio gives an indication of image quality, with $S = 1$ representing perfect (unaberrated) imaging conditions and decreasing values of S corresponding to increasing image degradation due to aberrations. For general imaging $S < 0.8$ is considered to represent poor imaging conditions.* For $\sigma = 1.10$ rad, this results in a Strehl ratio of 0.30. This is comparable to the Strehl ratio previously observed [46] when imaging at a depth of 10 μm in cleared *Drosophila melanogaster* brain tissue. The resulting SRRF images show more noise, no clear structure in the features highlighted in Fig. 6 b) and artefacts that appear to link previously distinct features. To demonstrate the effects of even higher aberrations, $\sigma = 1.43$, $S = 0.12$, we applied 3 units each of coma and astigmatism, with the results being shown in Fig. 7 e) and f). While SRRF still recovers some features with this level of aberration, the image is significantly degraded when compared to either of the other two aberration cases. *It is worth noting that the Strehl ratio can also be calculated as the ratio of the peak aberrated intensity to the peak unaberrated intensity. From the peak intensities in Fig. 7 a,c,e), and assuming that a) is a reasonable approximation of the no-aberration case, we find that the measured low aberration Strehl ratio is 0.54, while the high aberration is 0.13. We attribute the difference between the expected Strehl ratios and the empirical ones to be due to drift in the calibration of the deformable mirror. While Fig. 4 shows that the applied modes behave according to their expected aberration class, we did not obtain sufficient quantitative information to confirm the modal amplitude calibration of the DM. Once more, we would like to emphasise that the photostability of the ND emission allows us to obtain high quality and trustworthy data, such as that being discussed in this paragraph, that is invaluable in the alignment and characterisation of complex optical microscopy experiments.*

To better quantify the effective resolution of the SRRF images in all three cases shown in Fig. 7, we applied the Fourier Ring Correlation technique (FRC) [47]. This works on a pair of

Figure 7. Demonstration of the effects of aberration on TPE and SRRF imaging. Left column: Drift-corrected sum of 100 TPE frames with no (top row, peak counts 92000), Low (middle row, peak counts 50000) and High (bottom row, peak counts 12000) amounts of applied aberration. Right column: Corresponding SRRF images. All images at same scale, with 1 μm bar show in a) for reference. Linear colour scale applied to each image.

images that differ only in noise: at low spatial frequencies which are dominated by the (identical) structure of the sample there is a high degree, $FRC \approx 1$, of correlation, which decreases with increasing spatial frequency until reaching a point at which the images are dominated entirely by uncorrelated noise. The spatial frequency at which this crossover occurs, often referred to as the FRC resolution, is a measure of the maximum spatial frequency (minimum feature size) that still contains information in the images. Figure 8 shows the FRC curves derived from each aberration case, with the FRC value plotted against inverse pixels as the unit of spatial frequency. *To generate the FRC curve, we performed drift correction on the original TPE image sequences, deinterleaved the resulting odd and even frames and processed them in the SRRF algorithm.* The resulting pair of SRRF images was then used for the FRC algorithm. Our chosen SRRF parameters resulted in a pixel size of 11.4 nm, which can be used to convert from spatial frequency to resolution.

Figure 8. Fourier-Ring Correlation curves for SRRF data in Fig. 7. Horizontal scaling is inverse pixels, with a single pixel real-space size of 11.4 nm. Also shown is a resolution threshold curve based on a half-bit of information per pixel. The FRC resolutions given in the text refer to when each curve first crosses the threshold.

Also shown on Fig. 8 is the threshold curve used to determine the FRC resolution. Common approaches to determine the FRC resolution use a fixed value of $1/7$ as the threshold for the FRC resolution: the spatial frequency at which the FRC curve first falls under $1/7$ is given as the resolution. A problem with this is, for low spatial frequencies, the FRC curve is derived from Fourier-domain rings that contain fewer pixels than those for higher spatial frequencies. There is therefore a difference in the statistical behaviour in these two conditions [48,49] for which a better threshold is the curve defining the FRC level at which each pixel contains half of one bit of information. We include this "Half-bit Threshold" on Fig. 8 and use it to determine the resolution for each of our images. The resulting resolutions are given in Table 1

Aberration	SRRF FRC (Pixels ⁻¹)	SRRF FRC (nm)	TPE FRC (nm)
None	0.269	42.5	250
Low	0.235	49	262
High	0.062	184	304

Table 1. FRC derived resolution for images acquired under differing optical aberrations. SRRF data from Fig. 8, TPE data derived from FRC of the original image stacks used to generate the SRRF images.

That the FRC resolution agrees with the FWHM of the Gaussian fit in Fig. 6 b) is gratifying. *We note that the FRC measures self-similarity of images, and does so in a manner that averages across the entire image. As such, the number obtained from the FRC curve, and which is generally considered to be equivalent to the resolution, does not necessarily hold across the entire image. Indeed, despite the FRC resolution of 49 nm, the upper central pair of ND in Fig. 7 d) are not resolved. We therefore emphasise again that the FRC is a useful tool, but that there is still a widely acknowledged, and unsolved, problem of the definition of image resolution in super-resolution microscopy.* What is nevertheless surprising is the robustness of the SRRF algorithm to aberrations - we see evidence of resolution enhancement even in the case of strongly aberrated images, *although it is doubtful how much information can usefully be extracted from the SRRF image in the high aberration case.*

Nanodiamond photostability

To further demonstrate the exceptional photostability of ND, we prepared a sample of 70 nm ND (Sigma Aldrich) on a #1.5 coverslip using immersion oil as the mounting medium. Under 532 nm excitation we were able to image a single ND over multiple hours without any loss of signal. Figure 9 shows an example of this photostability; there is little change in the fluorescence detected over a period of 100 s of continuous excitation of a single ND using 1 mW of average power at 532 nm. Each data point was taken sequentially with a 2 ms integration time: this plot shows 5×10^4 fluorescence measurements. The small loss of signal by the end of the run is entirely due to sample drift and can be fully recovered by realigning the excitation laser to the ND location. Under TPE conditions we observed directly equivalent stability. By way of comparison with conventional fluorophores, with our microscope we typically observe a permanent decrease in fluorescence of 50% in samples labelled with Alexa 488 after 100 frames of 1.5 ms pixel duration and 100 μ W of excitation power. With twice-Nyquist sampling conditions, this implies an effective photobleaching half-life of 2.4 s. Thus, after an equivalent 100 s duration experiment to that shown in Fig. 9, the Alexa 488 would be emitting at an intensity 3×10^{-13} of which it began.

Conclusion

Nanodiamond has been shown to have much promise as a fluorophore for use in biological imaging. With this study we have shown that it is not only an exceptional fluorophore in single photon imaging, but that it is also superb when used with two-photon excitation microscopes. Furthermore, we demonstrate that its photostability gives it significant advantages when performing adaptive-optics or computational super-resolution images. Indeed, we saw evidence for a ten-fold resolution increase when generating SRRF images from our two-photon data.

Data accessibility statement

All data underpinning this publication, along with descriptions of the file format generated by the custom software running our microscope and scripts for importing the data into Fiji are openly

Figure 9. Demonstration of the photostability of ND under continuous single-photon excitation. Laser power was 1 mW at 532 nm using 40 MHz repetition rate. Sample points were generated consecutively with a 2 ms detector integration time.

available from the University of Strathclyde KnowledgeBase at <https://doi.org/10.15129/2365ed7a-4695-4928-b748-44308585b12>

Research ethics statement

The work in this paper does not require this statement

Animal ethics statement

The work in this paper does not require this statement

Permission to carry out fieldwork statement

The work in this paper does not require this statement

Funding statement

This work was funded under grants from the Royal Society (RGF\EA\181058 and URF\R\180017) and EPSRC (EP/M003701/1). GSC is funded under "OPTIMA: The EPSRC and MRC Centre for Doctoral Training in Optical Medical Imaging". BRP holds a Royal Society University Research Fellowship.

Competing interests statement

We have no competing interests to declare

Authors' contributions statement

GEJ constructed the microscope, participated in the data acquisition, performed data analysis, and drafted the manuscript; GSC constructed the microscope, performed data acquisition and

microscope calibration experiments; BRP conceived of the work, constructed the microscope, participated in the data acquisition, performed data analysis and helped draft the manuscript. All authors gave final approval for publication.

Acknowledgements

We wish to acknowledge the loan of a Fidelity 2 laser from Coherent Inc., which was used to generate all two-photon excitation data in this paper.

References

1. Ying Zhu, Jing Li, Wenxin Li, Yu Zhang, Xiaofeng Yang, Nan Chen, Yanhong Sun, Yun Zhao, Chunhai Fan, and Qing Huang.
The Biocompatibility of Nanodiamonds and Their Application in Drug Delivery Systems.
Theranostics, 2(3):302–312, 2012.
2. Mayeul Chipaux, Kiran J. van der Laan, Simon R. Hemelaar, Masoumeh Hasani, Tingting Zheng, and Romana Schirhagl.
Nanodiamonds and Their Applications in Cells.
Small, 14(24):1704263, June 2018.
3. Basem Moosa, Karim Fhayli, Song Li, Khatchatur Julfakyan, Alaa Ezzeddine, and Niveen M. Khashab.
Applications of nanodiamonds in drug delivery and catalysis.
Journal of Nanoscience and Nanotechnology, 14(1):332–343, January 2014.
4. Dae Gon Lim, Racelly Ena Prim, Ki Hyun Kim, Eunah Kang, Kinam Park, and Seong Hoon Jeong.
Combinatorial nanodiamond in pharmaceutical and biomedical applications.
International Journal of Pharmaceutics, 514(1):41–51, November 2016.
5. Yuen Yung Hui, Wesley Wei-Wen Hsiao, Simon Haziza, Michel Simonneau, François Treussart, and Huan-Cheng Chang.
Single particle tracking of fluorescent nanodiamonds in cells and organisms.
Current Opinion in Solid State and Materials Science, 21(1):35–42, February 2017.
6. Wesley Wei-Wen Hsiao, Yuen Yung Hui, Pei-Chang Tsai, and Huan-Cheng Chang.
Fluorescent Nanodiamond: A Versatile Tool for Long-Term Cell Tracking, Super-Resolution Imaging, and Nanoscale Temperature Sensing.
Accounts of Chemical Research, 49(3):400–407, March 2016.
7. H. L. Shergold and C. J. Hartley.
The surface chemistry of diamond.
International Journal of Mineral Processing, 9(3):219–233, July 1982.
8. Andreas Nagl, Simon Robert Hemelaar, and Romana Schirhagl.
Improving surface and defect center chemistry of fluorescent nanodiamonds for imaging purposes - a review.
Analytical and Bioanalytical Chemistry, 407(25):7521–7536, October 2015.
9. V. Vermeeren, L. Grieten, N. Vanden Bon, N. Bijmens, S. Wenmackers, S. D. Janssens, K. Haenen, P. Wagner, and L. Michiels.
Impedimetric, diamond-based immunosensor for the detection of C-reactive protein.
Sensors and Actuators B: Chemical, 157(1):130–138, September 2011.
10. Shabnam Siddiqui, Zhenting Dai, Courtney J. Stavis, Hongjun Zeng, Nicolaie Moldovan, Robert J. Hamers, John A. Carlisle, and Prabhu U. Arumugam.
A quantitative study of detection mechanism of a label-free impedance biosensor using ultrananocrystalline diamond microelectrode array.
Biosensors and Bioelectronics, 35(1):284–290, May 2012.
11. Wensha Yang, James E. Butler, John N. Russell, and Robert J. Hamers.
Interfacial Electrical Properties of DNA-Modified Diamond Thin Films: Intrinsic Response and Hybridization-Induced Field Effects.
Langmuir, 20(16):6778–6787, August 2004.
12. A. T. Collins, G. Davies, H. Kanda, and G. S. Woods.
Spectroscopic studies of carbon-13 synthetic diamond.

- Journal of Physics C: Solid State Physics*, 21(8):1363, 1988.
13. Tsai-Jung Wu, Yan-Kai Tzeng, Wei-Wei Chang, Chi-An Cheng, Yung Kuo, Chin-Hsiang Chien, Huan-Cheng Chang, and John Yu.
Tracking the engraftment and regenerative capabilities of transplanted lung stem cells using fluorescent nanodiamonds.
Nature Nanotechnology, 8:682, August 2013.
 14. Carlo Bradac, Torsten Gaebel, Chris I. Pakes, Jana M. Say, Andrei V. Zvyagin, and James R. Rabeau.
Effect of the Nanodiamond Host on a Nitrogen-Vacancy Color-Centre Emission State.
Small, 9(1):132–139, 2013.
 15. Yuen Yung Hui, Chia-Liang Cheng, and Huan-Cheng Chang.
Nanodiamonds for optical bioimaging.
Journal of Physics D: Applied Physics, 43(37):374021, September 2010.
 16. Shingo Sotoma, Daiki Terada, Takuya F. Segawa, Ryuji Igarashi, Yoshie Harada, and Masahiro Shirakawa.
Enrichment of ODMR-active nitrogen-vacancy centres in five-nanometre-sized detonation-synthesized nanodiamonds: Nanoprobes for temperature, angle and position.
Scientific Reports, 8(1):5463, April 2018.
 17. S. R. Hemelaar, P. de Boer, M. Chipaux, W. Zuidema, T. Hamoh, F. Perona Martinez, A. Nagl, J. P. Hoogenboom, B. N. G. Giepmans, and R. Schirhagl.
Nanodiamonds as multi-purpose labels for microscopy.
Scientific Reports, 7(1):720, April 2017.
 18. Silvia Arroyo-Camejo, Marie-Pierre Adam, Mondher Besbes, Jean-Paul Hugonin, Vincent Jacques, Jean-Jacques Greffet, Jean-François Roch, Stefan W. Hell, and François Treussart.
Stimulated emission depletion microscopy resolves individual nitrogen vacancy centers in diamond nanocrystals.
ACS nano, 7(12):10912–10919, December 2013.
 19. Gregoire Laporte and Demetri Psaltis.
STED imaging of green fluorescent nanodiamonds containing nitrogen-vacancy-nitrogen centers.
Biomedical Optics Express, 7(1):34–44, December 2015.
 20. Paul Delaney, James C. Greer, and J. Andreas Larsson.
Spin-Polarization Mechanisms of the Nitrogen-Vacancy Center in Diamond.
Nano Letters, 10(2):610–614, February 2010.
 21. V. M. Acosta, E. Bauch, M. P. Ledbetter, A. Waxman, L.-S. Bouchard, and D. Budker.
Temperature Dependence of the Nitrogen-Vacancy Magnetic Resonance in Diamond.
Physical Review Letters, 104(7):070801, February 2010.
 22. Sungkun Hong, Michael S. Grinolds, Linh M. Pham, David Le Sage, Lan Luan, Ronald L. Walsworth, and Amir Yacoby.
Nanoscale magnetometry with NV centers in diamond.
MRS Bulletin, 38(2):155–161, February 2013.
 23. Yuzhou Wu, Fedor Jelezko, Martin B Plenio, and Tanja Weil.
Diamond Quantum Devices in Biology.
Angewandte Chemie International Edition, 55(23):6586–6598, June 2016.
 24. A. Gruber, A. Dräbenstedt, C. Tietz, L. Fleury, J. Wrachtrup, and C. von Borczyskowski.
Scanning Confocal Optical Microscopy and Magnetic Resonance on Single Defect Centers.
Science, 276(5321):2012–2014, June 1997.
 25. Romana Schirhagl, Kevin Chang, Michael Loretz, and Christian L. Degen.
Nitrogen-vacancy centers in diamond: nanoscale sensors for physics and biology.
Annual Review of Physical Chemistry, 65:83–105, 2014.
 26. Fritjof Helmchen and Winfried Denk.
Deep tissue two-photon microscopy.
Nature Methods, 2(12):932–940, December 2005.
 27. Tse-Luen Wee, Yan-Kai Tzeng, Chau-Chung Han, Huan-Cheng Chang, Wunshain Fann, Jui-Hung Hsu, Kuan-Ming Chen, and Yueh-Chung Yu.
Two-photon Excited Fluorescence of Nitrogen-Vacancy Centers in Proton-Irradiated Type Ib Diamond.
The Journal of Physical Chemistry A, 111(38):9379–9386, September 2007.

28. Peng Ji, R. Balili, J. Beaumariage, S. Mukherjee, D. Snoke, and M. V. Gurudev Dutt. Multiple-photon excitation of nitrogen vacancy centers in diamond. *Physical Review B*, 97(13):134112, April 2018.
29. Chiara Mauriello Jimenez, Nikola Z. Knezevic, Yolanda Galàn Rubio, Sabine Szunerits, Rabah Boukherroub, Florina Teodorescu, Jonas G. Croissant, Ouahiba Hocine, Martina Seric, Laurence Raehm, Vanja Stojanovic, Dina Aggad, Marie Maynadier, Marcel Garcia, Magali Gary-Bobo, and Jean-Olivier Durand. Nanodiamond-PMO for two-photon PDT and drug delivery. *Journal of Materials Chemistry B*, 4(35):5803–5808, August 2016.
30. Yuen Yung Hui, Bailin Zhang, Yuan-Chang Chang, Cheng-Chun Chang, Huan-Cheng Chang, Jui-Hung Hsu, Karen Chang, and Fu-Hsiung Chang. Two-photon fluorescence correlation spectroscopy of lipid-encapsulated fluorescent nanodiamonds in living cells. *Optics Express*, 18(6):5896–5905, March 2010.
31. Yi-Ren Chang, Hsu-Yang Lee, Kowa Chen, Chun-Chieh Chang, Dung-Sheng Tsai, Chi-Cheng Fu, Tsong-Shin Lim, Yan-Kai Tzeng, Chia-Yi Fang, Chau-Chung Han, Huan-Cheng Chang, and Wunshain Fann. Mass production and dynamic imaging of fluorescent nanodiamonds. *Nature Nanotechnology*, 3(5):284–288, May 2008.
32. Siân Culley, Kalina L. Tosheva, Pedro Matos Pereira, and Ricardo Henriques. SRRF: Universal live-cell super-resolution microscopy. *The International Journal of Biochemistry & Cell Biology*, 101:74–79, August 2018.
33. Martin J. Booth. Adaptive optical microscopy: the ongoing quest for a perfect image. *Light: Science & Applications*, 3(4):e165, April 2014.
34. Martin Booth, Débora Andrade, Daniel Burke, Brian Patton, Mantas Zuraszkas. Aberrations and adaptive optics in super-resolution microscopy. *Microscopy*, 64(4):251–61, August 2015.
35. Xiaodong Tao, Andrew Norton, Matthew Kissel, Oscar Azucena, and Joel Kubby. Adaptive optical two-photon microscopy using autofluorescent guide stars. *Optics Letters*, 38(23):5075–5078, December 2013.
36. Delphine Débarre, Edward J. Botcherby, Tomoko Watanabe, Shankar Srinivas, Martin J. Booth, and Tony Wilson. Image-based adaptive optics for two-photon microscopy. *Optics Letters*, 34(16):2495–2497, August 2009.
37. M. a. A. Neil, R. Juškaitis, M. J. Booth, T. Wilson, T. Tanaka, and S. Kawata. Adaptive aberration correction in a two-photon microscope. *Journal of Microscopy*, 200(2):105–108, 2000.
38. Johannes Schindelin, Ignacio Arganda-Carreras, Erwin Frise, Verena Kaynig, Mark Longair, Tobias Pietzsch, Stephan Preibisch, Curtis Rueden, Stephan Saalfeld, Benjamin Schmid, Jean-Yves Tinevez, Daniel James White, Volker Hartenstein, Kevin Eliceiri, Pavel Tomancak, and Albert Cardona. Fiji: an open-source platform for biological-image analysis. *Nature Methods*, 9(7):676–682, June 2012.
39. Curtis T. Rueden, Johannes Schindelin, Mark C. Hiner, Barry E. DeZonia, Alison E. Walter, Ellen T. Arena, and Kevin W. Eliceiri. ImageJ2: ImageJ for the next generation of scientific image data. *BMC Bioinformatics*, 18(1):529, November 2017.
40. D A Green. A colour scheme for the display of astronomical intensity images. *Bulletin of the Astronomical Society of India*, 39:289, 2011.
41. K. Visscher, G. J. Brakenhoff, and T. D. Visser. Fluorescence saturation in confocal microscopy. *Journal of Microscopy*, 175(2):162–165, 1994.
42. Martin Booth, Tony Wilson, Hong-Bo Sun, Taisuke Ota, and Satoshi Kawata. Methods for the characterization of deformable membrane mirrors. *Applied Optics*, 44(24):5131–5139, August 2005.
43. A. Thayil and M. J. Booth.

- Self calibration of sensorless adaptive optical microscopes.
Journal of the European Optical Society - Rapid publications, 6(0), September 2011.
44. Martin J. Booth, Mark A. A. Neil, Rimas Juškaitis, and Tony Wilson.
 Adaptive aberration correction in a confocal microscope.
Proceedings of the National Academy of Sciences, 99(9):5788–5792, April 2002.
45. Romain F Laine, Kalina L Tosheva, Nils Gustafsson, Robert D M Gray, Pedro Almada, David Albrecht, Gabriel T Risa, Fredrik Hurtig, Ann-Christin Lindås, Buzz Baum, Jason Mercer, Christophe Leterrier, Pedro M Pereira, Siân Culley, and Ricardo Henriques.
 NanoJ: a high-performance open-source super-resolution microscopy toolbox.
Journal of Physics D: Applied Physics, 52(16):163001, feb 2019.
46. Brian R. Patton, Daniel Burke, David Oswald, Travis J. Gould, Joerg Bewersdorf, and Martin J. Booth.
 Three-dimensional STED microscopy of aberrating tissue using dual adaptive optics.
Optics Express, 24(8):8862–8876, April 2016.
47. Robert P. J. Nieuwenhuizen, Keith A. Lidke, Mark Bates, Daniela Leyton Puig, David Grünwald, Sjoerd Stallinga, and Bernd Rieger.
 Measuring image resolution in optical nanoscopy.
Nature Methods, 10(6):557–562, June 2013.
48. Marin van Heel and Michael Schatz.
 Fourier shell correlation threshold criteria.
Journal of Structural Biology, 151(3):250–262, September 2005.
49. Marin van Heel and Michael Schatz.
 Reassessing the Revolution’s Resolutions.
bioRxiv, page 224402, November 2017.